# Research on the Migration of the Total Manganese during the Process of Water Icing

**Yan Zhang \*, Yuanqing Tang, Aixin Yu, Wanli Zhao and Yucan Liu**

College of Civil Engineering, Yantai University, Yantai 264000, China
\* Correspondence: zhangyan-992@163.com; Tel.: +86-135-6259-1712

**Abstract:** Our research focused on the migration law of the total manganese (TMn) during the process of water icing. We utilized two experimental methods: (1) natural icing and (2) simulated icing. While using laboratory simulation, we explored the effects of ice thickness, freezing temperature, and initial concentrations on the migration of TMn in the ice-water system. The distribution coefficient "K" (the ratio of the average concentration of TMn in the ice body to the average concentration of TMn in the under-ice water body) was used to characterize it. The results indicated that TMn continuously migrated from ice to under-ice water during the process of water icing. The concentration of TMn in the ice was the upper layer < middle layer < lower layer, and K decreases as the ice thickness, freezing temperature, and initial concentration increased. We explained the migration of TMn during the process of water icing from the perspective of crystallography. Our research can arouse other researcher's attention towards the change of TMn concentration in lakes in high latitudes during the icebound period.

**Keywords:** natural icing; simulated icing; migration law; distribution coefficient "K"; TMn

## 1. Introduction

The icebound period is an important hydrological feature of surface water bodies in high-latitude regions [1]. Approximately half of all lakes in the world (more than 50 million) are regularly frozen [2]. The lakes, reservoirs, and ponds are covered with ice and snow during the icebound period, which reduced the penetration of light [3,4]. When the thickness of snow reached 10 cm, light penetration was reduced to level that was insufficient for photosynthesis [5]. The ice cover prevented external water from entering the under-ice water, weakened the exchange of greenhouse gases between the atmosphere and the water, and weakened the exchange of energy between them [6]. This decreased the rate of re-oxygenation [7–9], dilution [10], photolysis [11], and various biochemical reactions [12,13]. Therefore, the efficiency of water body's self-purification declined, and the migration characteristic of substance has its particularity during the icebound period [14,15].

However, there are few studies on the migration of substances in ice-water systems during the icebound period [16], only 2% of the literatures on freshwater body refer to the water icing process [17]. Belzile et al. [18] studied the dissolution of colored organics in polar and alpine lakes; those studies found that there are only a small number of low-molecular-weight pollutants (dissolved organic carbon and colored dissolved organic matter) in ice bodies and that most of the pollutants are discharged into the under-ice water body. Hampton et al. [11] conducted the first worldwide quantitative analysis of 101 lakes (mainly located in the northern hemisphere) and found that TDN (total dissolved nitrogen) and TN (total nitrogen) were higher in the under-ice water during the icebound period. Powers et al. [19] found that the concentration of $NO_3$-N (nitrate nitrogen) in the under-ice water gradually increased during the icebound period, showing a strong positive correlation with the number of frozen days. Roger and Gregory [20] found that approximately 99% of salts migrated into the under-ice water during

the process of water icing while using the on-site observations of meteorological and environmental indicators at Tailings Lake in northwestern Canada. Most of the existing researches focused on the migration of organic pollutants and nutrients during the process of natural icing, but there are few studies on the migration laws and mechanisms of metal substance in the icing process. In this study, San yuan lake was selected as the research object. The concentration of the total manganese (TMn) in San yuan lake exceeded 0.3 mg/L during the non-frozen period, which was much higher than the limit of Class III water according to Chinese Standard of Surface Water Environmental Quality (GB3838-02) (the limit concentration of TMn ≤ 0.1 mg/L) [21]. TMn was one of the typical pollutants, which caused the water body to produce a peculiar smell and affect the environment [22]. However, the concentration of TMn may increase further during the icing process, which may cause varying degrees of harm to the human nervous system, reproductive system, vital organs, and cardiovascular system [23–27]. Moreover, TMn circulation in water affects the migration of other trace metal elements [28], while the migration and cycling of TMn may have its own particularities during the icebound period. Accordingly, revealing is the interface circulation of TMn, which helps to further our understanding of element circulation and water source protection.

The purpose of this study is to investigate the distribution of TMn in the ice-water system during the natural icing process; the effects of temperature, thickness, and initial concentration on the distribution of TMn were investigated in the simulated icing experiment; the distribution coefficient K was used to characterize the migration ability of TMn, which is to arouse other researcher's attention to the changes of surface water TMn pollution during the icebound period.

## 2. Materials and Methods

### 2.1. Natural Icing Experiment

San yuan Lake, located in Yan tai, Shandong Province (37°25′ N, 121°34′ E), was selected to study the migration of TMn during the process of natural icing. We selected a sub-area of 1200 m² as the sampling point, with an average depth of 1.5 m, and no inflow or outflow except for rainfall and evaporation. Generally, the icebound period is from November to February of the next year and the average thickness of the ice over many years is 18 cm. In November of 2017, water sample was collected 0.5 m below the surface of the water during the non-frozen period. In December of 2017, ice sample with a thickness of 5 cm and water 0.5 m below the ice was collected during the icebound period. In January of 2018, an ice sample with a thickness of 15 cm and water 0.5 m below the ice was collected during the icebound period, sampling principles according to the 'Water and Wastewater monitoring analysis method' [29]. The water samples were stored in polyethylene bottles that were acidified with nitric acid to pH < 2, and the ice samples were stored in a cryostat and brought back to the laboratory. These ice samples were divided into three equal parts: the upper, middle, and lower layers. Moreover, the upper, middle, and lower layers samples were melted under room temperature conditions.

In this study, the TMn content was measured in water samples and the ice-melt water samples. The water samples were not filtered, and 25 mL of well-mixed water samples were placed in 50 mL colorimetric tube. We adjusted the pH to a neutral level with (1 + 9) ammonia water, added 10 mL of potassium pyrophosphate-sodium acetate buffer solution, shook well, then added 3 mL of 2% potassium periodate solution, diluted to the mark with water, and then shook well again. After standing for 10 min., the absorbance was measured with a 50 mm cuvette at a wavelength of 525 nm while using water as a reference. The TMn content in the water sample was calculated based on the obtained absorbance and calibration curve. For all samples, the concentration of TMn was tested in triplicate, according to the 'Water and Wastewater monitoring analysis method' [29] and Sun's study [30]; the standard deviation of sample test result was controlled within 5%.

### 2.2. Simulated Icing Experiment

#### 2.2.1. Icing Simulation Device

Our study utilized an open unidirectional downward icing simulator in order to simulate the top-down icing process of natural water bodies (Figure 1). A glass cylindrical barrel was wrapped with Expanded Polystyrene (EPS) insulation to block the transfer of heat between the barrel and the outdoor. A temperature-controlled heating sheet was placed between the outer part of the barrel and the thermal insulation to facilitate the removal of the ice sample. A resistance wire (nickel-chromium alloy) was placed in the barrel to measure ice thickness. This device was then placed in a low-temperature testing box. The lowest temperature that could be attained in the testing box was −40 °C and the temperature fluctuation did not exceed 0.5 °C.

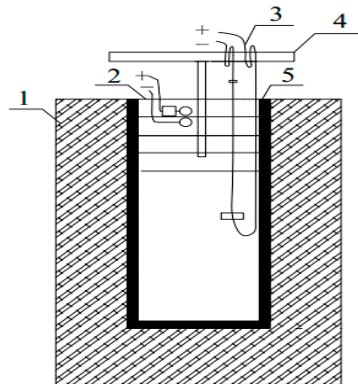

**Figure 1.** Icing Simulation Device (**1**) Expanded Polystyrene (EPS) insulation. (**2**) Heating sheet. (**3**) Device for measuring ice thickness. (**4**) T-bracket. (**5**) Glass cylindrical barrel.

#### 2.2.2. Experimental Method

The simulated icing experiment includes the following three parts:

(1) We prepared the TMn standard solution with the concentration of 0.22 mg/L to study the effect of the freezing thickness on the migration of TMn in the process of water icing, which was placed in three icing simulation device. Subsequently, we placed 8 L water sample in each icing simulation device and placed the devices in the low-temperature testing box at −10 °C. The samples were removed when the thickness of the ice reached 5 cm, 10 cm, and 15 cm.

(2) We prepared the TMn standard solution with the concentration of 0.22 mg/L to study the effect of the temperature on the migration of TMn in the process of water icing, which was placed in three icing simulation device. Afterwards, we placed 8L water sample in each icing simulation device and placed the devices in the low-temperature testing box at −10 °C, −15 °C, and −20 °C. The samples were removed when the thickness of the ice reached 10 cm.

(3) To study the effect of initial concentration on the migration of TMn in the process of water icing. According to 'Standards for drinking water quality' [31], the concentration of TMn in drinking water shall not exceed 0.1 mg/L, and the concentration of TMn exceeding 0.3 mg/L will cause peculiar smell in water. Therefore, the concentration of standard solution of TMn is set as 0.11 mg/L, 0.22 mg/L, and 0.44 mg/L. We prepared TMn standard solutions with concentration of 0.11 mg/L, 0.22 mg/L, and 0.44 mg/L in three icing simulators. Subsequently, we placed 8 L water sample in each icing simulation device, and then placed the devices in the low-temperature testing box at −10 °C. The samples were removed when the thickness of the ice reached 10 cm. TMn standard storage solution (1000 mg/L): we took 0.500 g electrolytic manganese (99.9%) and dissolved it in 10 mL (1 + 1) nitric acid, and transferred it to 500 mL volumetric bottle after heating and dissolving. The manganese standard solution in the simulated freezing experiment was obtained by diluting the above manganese standard storage solution (1000 mg/L).

The simulated icing experiments were single-factor experiment, so the different concentrations of manganese standard solution were configured with deionized water. Additionally, the ice samples that were obtained from the icing simulation device were divided into three layers: upper, middle, and lower. Each sample was placed in a beaker and melted at room temperature. The under-ice water was evenly mixed and then removed and placed in a beaker. The TMn of the samples was detected by the same method as that of the samples during the natural freezing process. For all of the samples, the concentration of TMn was tested in triplicate, the standard deviation of sample test results was controlled within 5%. In each group, the ice layer that was connected to the cold source at the upper part of the ice body was defined as the upper layer, the ice layer in the middle of the ice body was defined as the middle layer, and the ice layer at the bottom of the ice body connected to the under-ice water body was defined as the lower layer.

The variation range of the pH for the lake was 6.9–7.6 during the non-frozen and icebound period of San yuan lake, with an average value of 7.25 and the coefficient of variation of 4.8%. The variation range of the pH for the San yuan lake was small during the non-frozen and icebound period, and the influence of pH on the distribution coefficient is very limited.

The variation range of the DO (dissolved oxygen) for the lake was 7.2–12.5 mg/L during the non-frozen period and the icebound period of San yuan lake, with an average value of 9.8 mg/L and the coefficient of variation of 21.2%. As the variation range of the pH and DO for the San yuan lake was small during the non-frozen and icebound period, and the simulated icing experiment did not explore the effect of pH and DO on the migration law of TMn. In the simulated icing experiment, we adjust pH with sodium hydroxide solution, the pH range of the standard solution was 6.5–7.0, and the DO range was 9.5–15.5 mg/L. We used Multiparameter tester (USA ORION VERSA STAR) to measure the pH and DO of the water samples.

The distribution coefficient (K) was the ratio of the average concentration of TMn in the ice body to the average concentration of TMn in the under-ice water body. It reflected the ability of TMn to migrate into under-ice water during water freezing.

$$K = C_i/C_w$$

In this formula, $C_i$ was the average concentration of TMn in the ice and $C_w$ was the average concentration of TMn in the under-ice water body.

## 3. Results and Discussion

### 3.1. Natural Icing Experiment

The average concentration of TMn in the lake during the non-frozen period was 0.33 mg/L. As shown in Figure 2, when the ice thickness reached 5 cm, the concentration of TMn in the upper, middle and lower layers was 0.14 mg/L, 0.15 mg/L, and 0.21 mg/L, respectively. The average concentration of TMn in the ice was 0.17 mg/L and the concentration of TMn in the under-ice water was 0.42 mg/L, resulting K = 0.405. The concentration of TMn in under-ice water was 0.27 times higher than that of before freezing and 1.47 times higher than that of ice body.

As shown in Figure 2, when the ice thickness reached 15 cm, the concentration of TMn in the upper, middle, and lower layers was 0.12 mg/L, 0.16 mg/L, and 0.20 mg/L, respectively. The average concentration of TMn in the ice was 0.16 mg/L and the concentration of TMn in the under-ice water was 0.45 mg/L, resulting K = 0.356. The concentration of TMn in under-ice water was 0.36 times higher than that of before freezing and 1.81 times higher than that of ice body. However, with the increase of ice thickness, the temperature continued to decrease in winter, and the change of ice growth rate was not obvious, so there was no obvious migration of manganese in the ice body.

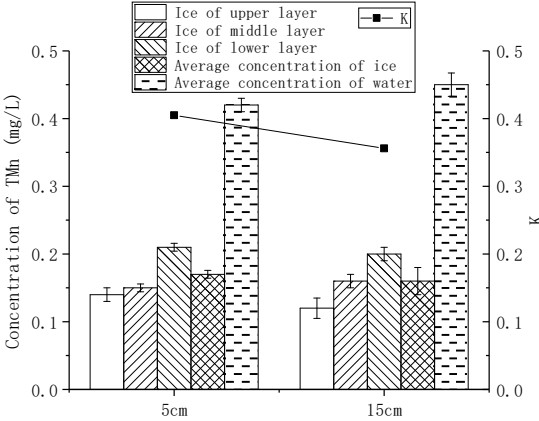

**Figure 2.** Distribution of total manganese (TMn) in Ice-water System During Natural Icing.

The concentration of TMn increased with ice thickness accordingly. During the natural icing process, Mn and its compounds are easily absorbed by organisms and gradually accumulated through the food chain. Mn has a strong affinity with many enzymes and active groups, which inhibits the activity of enzymes and hinders the metabolism of the body [32]. Studies have shown that the high concentration of Mn inhibits plant absorption of ions, such as $Ca^{2+}$, $Fe^{2+}$, and $Mg^{2+}$, inhibits plant activity, and causes oxidative stress, leading to oxidative damage, resulting in decreased chlorophyll and Rubisco content, destruction of chloroplast ultrastructure and decreased photosynthetic rate [33]; the increase in the concentration of Mn in the under-ice water should be of concern.

In summary, the distribution of TMn in ice-water system before and after lake freezing was expressed as: ice body < pre-freezing water body < under-ice water body. In other words, TMn migrated from the ice body to the under-ice water body during the process of water icing. These results were consistent with those of Sun [30], who confirmed the mechanism for the migration of heavy metal ions from ice to water when lakes are freezing from the perspective of energy change. As the thickness of ice increased, the concentration of TMn in the under-ice water body increased continuously, while K continued to decrease, the process of water icing caused the migration of Mn into the under-ice water.

The migration of TMn can be explained from the perspective of crystallography in the process of water icing. As the water temperature decreased, the diffusion effect of water molecules weakened and the degree of disorder in the system decreased. When the water temperature dropped to freezing, the water molecules aggregated to produce ice cores. The system spontaneously produced ice crystals when the water temperature dropped below freezing. During the super-cooling process, the random fluctuation that was caused by the precipitation of some ions in the water body and the pressure change of the ice water system increased the possibility of forming a critical size [34,35]. In turn, this promoted the heterogeneous nucleation of the water body to accelerate the formation of ice. The growth rate of ice crystals was related to the rate at which water molecules were added to the ice cores and the state of the solid-liquid interface. Water molecules that were close to the interface adhered to the surface of the ice cores through interface transitions [36]. Near the solid-liquid interface, the water molecules were precipitated by hydrogen bonding and adhered to the bottom of the ice. At the same time, TMn were squeezed out and escaped to the water body under the ice [37]. Thus, near the solid-liquid interface, the concentration of water molecules was much lower than that of water molecules in the whole liquid phase, and the concentration of TMn was much higher than that of the TMn in the whole liquid phase. Being driven by the difference in concentrations (Figure 3), water molecules in the liquid phase diffused to the solid-liquid interface, and TMn in the solid-liquid interface diffused to the liquid phase [38]. That is, TMn migrated from the ice body to the water body in the process of water icing. Therefore, the concentration of TMn in the ice body was lower than that in the under-ice water body.

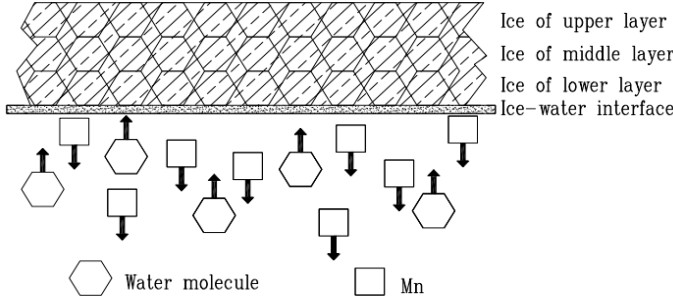

**Figure 3.** Migration of Water Molecules and Mn.

## 3.2. Simulated Icing Experiment

### 3.2.1. Effects of Ice Thickness on the Migration of TMn

As shown in Figure 4, when the ice thickness reached 5 cm, 10 cm, and 15 cm, the average concentration of TMn in the ice body was 0.18 mg/L, 0.11 mg/L, and 0.08 mg/L, respectively. The average concentration of TMn in the under-ice water body was 0.38 mg/L, 0.39 mg/L, and 0.51 mg/L, respectively. K was 0.474, 0.282, and 0.157, respectively. In other words, as the ice thickness increased, the K decreased, and the ability of the TMn to migrate into under-ice water increased as the ice thickness increased.

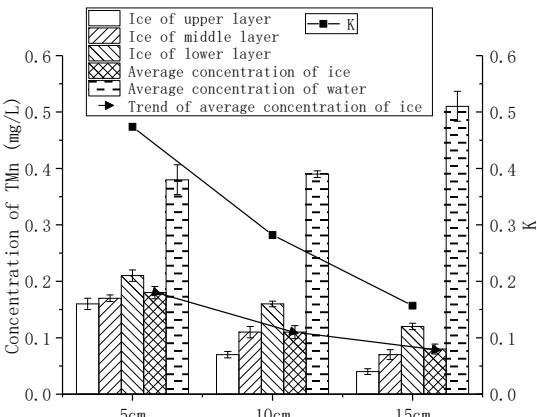

**Figure 4.** Distribution of TMn in the Ice-water System. Under Different Conditions of Ice Thickness.

As the freezing thickness increased, the average TMn content of each layer and body of ice decreased, but the rate of decline gradually decreased (Figure 4, shows the trend of average concentration of ice), which was consistent with the relationship between the Antarctic sea ice salinity and ice thickness that were studied by Fedotov [39]. This was because, in the early phase of icing, TMn trapped in the ice body continued to migrate to the under-ice water through the pore channels [40]. With the thickness of the ice increased, the rate of heat exchange between the under-ice water and the outside weakened. The rate of growth of ice was reduced [41], which resulted in the volume of newly formed ice crystals became larger, but the ability to capture manganese decreased, and there were fewer channels for discharging TMn in the ice per unit area, which were converted into a series of Mn cells and air cells [42]. The average concentration of TMn in the ice body gradually decreased as the thickness of the ice increased. In the three experiments, K reached a minimum when the ice thickness was 15 cm.

### 3.2.2. Effects of Freezing Temperatures on the Migration of TMn

As shown in Figure 5, when the freezing temperature was −10 °C, −15 °C, and −20 °C, the average concentration of TMn in the ice was 0.11 mg/L, 0.13 mg/L, and 0.15 mg/L, respectively. The average concentration of TMn in the under-ice water was 0.39 mg/L, 0.35 mg/L, and 0.30 mg/L, respectively. K decreased as the freezing temperature increased, with values of 0.282, 0.371, and 0.500, respectively.

That is, the lower the freezing temperature, the less TMn migrated into under-ice water. Therefore, there were more TMn trapped in the ice body.

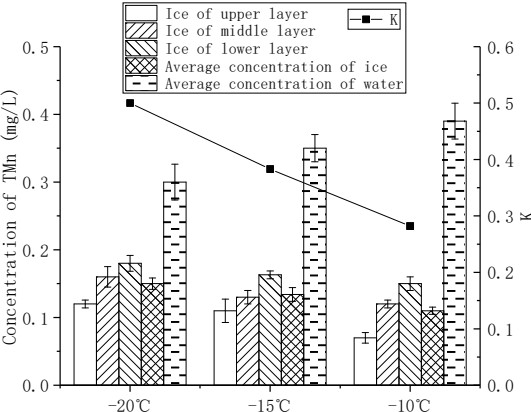

**Figure 5.** Distribution of TMn at Varying Freezing Temperatures in the Ice-water System.

As the lower the temperature was, the faster the ice crystal growth rate. Ice crystals formed finer and denser branches, capturing more TMn in the ice body; on the other hand, water molecules moved faster toward the solid-liquid interface. Once that speed exceeded the speed at which TMn moved toward the interface, TMn were trapped by the ice crystals [43]. Schmidt et al. also showed that the effect on the removal of pollutants was evident when the growth rate of ice was low [44,45]. Waller and Terwilliger [46,47] studied the effect of the freezing rate on the discharge of salt from the ice body. They found that the discharge of salt was affected by the thickness of the salt boundary at the ice-water interface. Further, the boundary thickness was inversely proportional to the rate of freezing. The lower the freezing temperature, the higher the freezing rate, which leads to the formation of more brine inclusions in the ice, which is the same as in this experiment.

### 3.2.3. Effects of Initial Concentrations on the Migration of TMn

As shown in Figure 6, when the initial concentrations were 0.11 mg/L, 0.22 mg/L, and 0.44 mg/L, the average concentration of TMn in the ice was 0.07 mg/L, 0.11 mg/L, and 0.18 mg/L, respectively. The average concentration of TMn in the under-ice water was 0.22 mg/L, 0.39 mg/L, and 0.75 mg/L, respectively. K was 0.318, 0.282, and 0.240, respectively. That is, the greater the initial concentration of TMn in the non-freezing period, the greater the concentration of TMn in the ice after icing, and the decrease in K with the initial concentration. This was consistent with the experimental results of W. Gao [48].

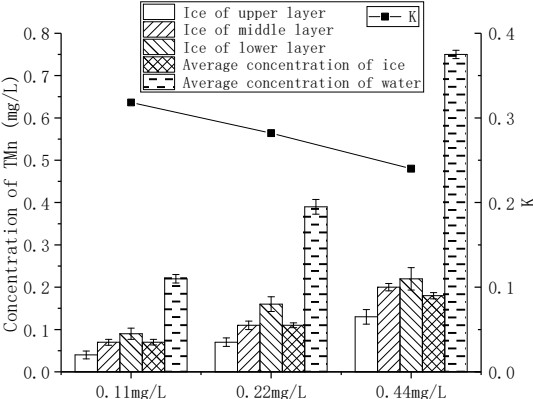

**Figure 6.** Distribution of TMn of Varying Initial Concentrations in the Ice-water System.

As the initial concentration of TMn increased, the viscosity coefficient increased, and the diffusion coefficient decreased. The probability of collision of ice crystals increased, and the TMn were more likely to be trapped when the ice grew, so that more TMn remained in the ice body [49,50]. As the initial concentration of TMn increased, more TMn was trapped in the ice during the initial icing period. However, with the increase of ice thickness, the total amount of TMn migrated from the ice body to the under-ice water increased. The increase of TMn in the ice body is much smaller than that in the under-ice water, which resulted in the decrease of K. The concentration distribution of TMn in the ice was: the upper layer < the middle layer < the lower layer, these results were similar to those of Weeks [51]. The concentration of TMn in the water body was the lowest during the non-frozen period, as the ice layer formed and the thickness of the ice increased, the TMn migrated to the under-ice water, which resulted in an increase in the concentration of TMn in under-ice water. Although the growth rate of the ice body slows down with the increase of ice thickness [41], the concentration of TMn trapped in the ice body is higher due to the increase of the concentration of TMn in the under-ice water. Further, the TMn continuously migrated downward in the pore channel under the influence of gravity. Additionally, the spatial density of the pore channel decreased with freezing time, until it finally disappeared [51]. The un-discharged TMn were trapped at the bottom of the pore channels, causing the concentration of TMn in the vertical direction of the ice body to increase with the increase of ice thickness.

## 4. Conclusions

Based on the results of the experiment, the following conclusions were drawn:

(1) Natural icing and simulated icing experiments show that: the concentration of TMn in the ice body < the concentration of TMn in the water body before icing < the concentration of TMn in the under-ice water body after icing. In the natural icing experiment, the migration of TMn leads to the further aggravation of TMn pollution in San yuan lake. Accordingly, we should pay more attention to the TMn pollution during the icebound period. Additionally, this study explains the migration of TMn in ice-water systems from the perspective of crystallography.

(2) In both natural icing and simulated icing experiments, the concentration of TMn in the ice body is shown as: the upper layer < the middle layer < the lower layer, K decreases with the increase of ice thickness, which indicated that the simulated icing experiment is consistent with the manganese migration law during the natural icing process.

(3) The simulated icing experiment shows that K decreases with the increase of icing temperature, icing thickness, and initial concentration. That is, higher icing temperature, larger icing thickness, and higher initial concentration are conducive to the migration of TMn to under-ice water bodies. Other factors that may affect the migration of TMn also need further explore.

## 5. Implication and Future Research

There are many forms of Mn in water, and the migration rules of Mn of different forms in the icing process may be different. In the future, we will study the migration rules of soluble Mn and particulate Mn.

The distribution of TMn in natural water is affected by various factors, such as organic ligand [52–55], pH [56], dissolved oxygen [57], and other factors. In the simulated icing experiment, this study only considered the effects of the freezing thickness, freezing temperature, and initial concentration on the migration distribution of TMn. In the future, it will be carried out that studies on the effects of organic ligands, pH, dissolved oxygen, mixing, light, freeze-thaw, and other factors on the migration of TMn.

**Author Contributions:** Data curation, Y.T., A.Y., W.Z., and Y.L.; Formal analysis, Y.T.; Methodology, A.Y., W.Z. and Y.L.; Software, Y.T. and A.Y.; Writing—original draft, Y.T.; Writing—Review & editing, Y.Z.

**Funding:** This work was supported by the National Natural Science Foundation of China (No.51609207), Key Research and Development Program (2019GHY112033) and the Foundation for Outstanding Young Scientist in Shandong Province (No.BS2014HZ021).

**Acknowledgments:** We are grateful for assistance with Yantai University Civil Engineering Laboratory, China.

**Conflicts of Interest:** The authors declare no conflict of interest.

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
