# Peer review of "Research on the Migration of the Total Manganese during the Process of Water Icing"

_water, doi:10.3390/w11081626_

Round 1
Reviewer 1 Report
Article: Research on the Migration of the Total Manganese uring the Process of Water Icing
Authors: Yan Zhang,, Yuanqing Tang, Aixin Yu, Wanli Zhao, Yucan Liu
General overview:
The article introduces the results of research concerning mechanism of migration Mn ions from ice to water. The article may be interesting for specialists in the field of aquatic ecology.
But I have some questions and comments to authors.
It is necessary to uniform the name of coefficient K (the ratio of the average concentration of TMn in the ice body to the average concentration of TMn in the under-ice water body): line 12 - the partition coefficient, line 57, 145 - the distribution coefficient K;
Line 75-84 - it is necessary to give citation to the English version of the method of Mn detection. There is International Standard ISO 6333:1986(2015) Water quality -- Determination of manganese -- Formaldoxime spectrometric method. Why authors didn’t use it?
Line 112. What kind of water was used in model experiments? Deionized or natural lake water?
Line 137. DO - it’s necessary to identify abbreviation and give the method of detection.
Figure 2, 4. There is not color visual difference between columns average concentration of water and ice of upper layer.
Did authors use statistic methods for identification the significance of difference between data in columns? Looking on the figures, it does hardly make a conclusion about the influence of ice thickness on the migration process of Mn.
Figure 2, 4. Why there is difference between data on fig.2 and fig. 4 for ice thickness 5 cm and 15 cm? Need to clarify, may be the conclusion about the absence of DO influence (line 141) is wrong?
Reviewer 2 Report
See attached file

Round 2
Reviewer 1 Report
Thank you for explanations and additions. I've got responses on my questions and remarks.
But I consider that additionally it will be better to give the reference on English (not Chines) with description of Mn determination in water. It will be appropriate for wide circle of readers.
